# Quantifying PD-L1 Expression to Monitor Immune Checkpoint Therapy: Opportunities and Challenges

**DOI:** 10.3390/cancers12113173

**Published:** 2020-10-29

**Authors:** Sridhar Nimmagadda

**Affiliations:** 1The Russell H. Morgan Department of Radiology and Radiological Science, Johns Hopkins University School of Medicine, Baltimore, MD 21287, USA; snimmag1@jhmi.edu; Tel.: +1-410-502-6244; Fax: +1-410-614-3147; 2Division of Clinical Pharmacology, Department of Medicine, Johns Hopkins University School of Medicine, Baltimore, MD 21287, USA; 3Department of Pharmacology and Molecular Science, Johns Hopkins University School of Medicine, Baltimore, MD 21205, USA; 4Bloomberg–Kimmel Institute for Cancer Immunotherapy, Johns Hopkins University School of Medicine, Baltimore, MD 21287, USA; 5The Sidney Kimmel Comprehensive Cancer Center, Johns Hopkins University School of Medicine, Baltimore, MD 21287, USA

**Keywords:** immune checkpoints, PET imaging, tumor microenvironment, immuno-Oncology, tumor mutational burden, interferon-γ signaling

## Abstract

**Simple Summary:**

Malignant cells hijack the regulatory roles of immune checkpoint proteins for immune evasion and survival. Therapeutics blocking those proteins can restore the balance of the immune system and lead to durable responses in cancer patients. Although a subset of patients derive benefit, there are few non-invasive technologies to guide and monitor those therapies to improve success rates. This is a review of the advancements in non-invasive methods for quantification of immune checkpoint protein programmed death ligand 1 expression, a biomarker detected by immunohistochemistry and widely used for guiding immune checkpoint therapy.

**Abstract:**

Therapeutics targeting programmed death ligand 1 (PD-L1) protein and its receptor PD-1 are now dominant players in restoring anti-tumor immune responses. PD-L1 detection by immunohistochemistry (IHC) is emerging as a reproducible biomarker for guiding patient stratification for those therapies in some cancers. However, PD-L1 expression in the tumor microenvironment is highly complex. It is upregulated by aberrant genetic alterations, and is highly regulated at the transcriptional, posttranscriptional, and protein levels. Thus, PD-L1 IHC is inadequate to fully understand the relevance of PD-L1 levels in the whole body and their dynamics to improve therapeutic outcomes. Imaging technologies could potentially assist in meeting that need. Early clinical investigations show promising results in quantifying PD-L1 expression in the whole body by positron emission tomography (PET). Within this context, this review summarizes advancements in regulation of PD-L1 expression and imaging agents, and in PD-L1 PET for drug development, and discusses opportunities and challenges presented by these innovations for guiding immune checkpoint therapy (ICT).

## 1. Introduction 

Tumors evolve to foster an immunosuppressive microenvironment and evade the immune system’s innate ability to recognize and eliminate malignant cells. To induce immunosuppression, tumors co-opt immune checkpoint proteins that otherwise regulate central and peripheral tolerance [1]. Monoclonal antibody (mAb) therapeutics that inhibit those immune checkpoints have resulted in impressive clinical success in a variety of cancers with durable survival rates [2]. However, most patients receiving ICT do not derive long-term benefits [3], necessitating the development of tools and techniques to better understand the immune effects of therapy [3]. 

PD-L1 and its receptor PD-1 have emerged as key cancer therapeutic targets amongst the many immune checkpoints that regulate host immunity [1]. PD-1 is an inhibitory receptor that is rapidly induced on naïve T cells to counteract T cell activation. PD-1 plays important roles in regulating T cell activation, tolerance, and exhaustion, and effector T cell responses [4]. PD-1 is expressed by several immune cell subtypes, including CD4+ and CD8+ T cells, regulatory T cells, B cells, and natural killer (NK) cells. PD-1 mediates regulatory functions through interaction with two ligands, PD-L1 and PD-L2. Many types of immune cells express PD-L1, including T cells, B cells, macrophages, dendritic cells (DCs), epithelial cells, stromal cells, endothelial cells, and tumor cells [4]. PD-L2 possesses high affinity for PD-1, and is expressed by DCs, macrophages, B cells, Th2 cells, and some lung epithelial cells. The interaction between PD-1 and its ligands leads to immunological ignorance in normal tissues and tumors. PD-1 ligands are induced in tissues or on tumor cells by proinflammatory stimuli. Those ligands then dampen PD-1 expressing effector T cell responses via a negative feedback mechanism, promote “adaptive immune resistance,” and protect tumors from immune attack [5]. PD-1 engagement with its ligands leads to diminished downstream signaling mediated through PI3K-AKT and Ras-MEK-ERK pathways, mitigation of T cell activation, and reduced killing capacity [3]. The observation that the majority of PD-1 activity is mediated through the widely expressed PD-L1 has led to the development of therapeutics targeting the PD(L)-1 axis, including multiple agents receiving FDA approval [3,6,7].

Three PD-1 (nivolumab, pembrolizumab, and tremelimumab) and three PD-L1 (atezolizumab, avelumab, and durvalumab) targeting monoclonal antibody (mAb) therapeutics have been approved as monotherapies or as therapies in combination with other immune checkpoint inhibitors, chemotherapeutics, or radiation. Comprehensive analyses of the clinical evaluation, therapeutic outcomes, pharmacokinetics (PK), and pharmacodynamic (PD) relationships of those antibodies have been extensively reviewed [6,7,8]. Given the success of those therapeutics, many candidate drugs and therapy combinations, in different development stages, are in need of robust biomarkers for therapy response assessment [9]. Although PD-L1 IHC is an FDA approved biomarker and remains the backbone of ICT development [3,8] significant efforts have also been made in identifying other biomarkers for response assessment [10,11,12]. Here, emphasis is placed on assessing the role of non-invasive quantification of PD-L1 levels in guiding future ICT development. 

## 2. Regulation of PD-L1 in the Tumor Microenvironment 

PD-L1-directed resistance to ICT is regulated by two general mechanisms [13]. In the tumor microenvironment (TME), innate immune resistance is driven by constitutively expressed PD-L1 on tumor cells that is mediated by genetic alterations (gene amplification [14], translocations [15], and disruption of 3′ UTR region [16]), aberrant oncogenic signaling, and hypoxia through hypoxia-inducible factor-1α [17]. PD-L1 expression in the TME is induced by activation of different oncogenic signaling pathways: in T cell lymphoma by ALK/STAT3 axis [18], in classical Hodgkin lymphoma by AP-1/JAK/STAT [19], in non-small-cell lung cancer (NSCLC) by EGFR [20], in BRAF inhibitor-resistant melanoma by c-jun/STAT3 [21], and in glioma by PI3K [22] to name a few. Myc, an oncogene that is aberrantly upregulated in a variety of cancers and tumorigenic [23], also mediates innate and adaptive immune resistance through the gene encoding for PD-L1 protein [24]. Major regulators of PD-L1 are summarized in Table 1 and were comprehensively reviewed by others [25].

PD-L1 expression is also induced in response to inflammatory stimuli on tumor cells, stromal cells, endothelial cells, T cells, APCs, and myeloid cells, and promotes adaptive immune resistance [26,27,28,29]. While tumor cell-derived PD-L1 is established as a requirement for neutralizing T-cell activity, recent studies suggest that host-derived immune cell PD-L1 is also essential for the PD(L)-1 blockade-mediated response [30,31,32]. It is possible that inherent immunogenicity of tumors could determine the relative contributions of these compartments in defining anti-tumor responses. Further studies are needed to delineate the relative contributions of tumor and host derived PD-L1 to anti-tumor immunity. Inflammatory signaling also regulates PD-L1 levels in the TME [27]. The predominant cytokine involved in upregulation of PD-L1 is IFNγ, which is secreted by immune cells homed to the TME during an antitumor immune response [5,29]. IFN-γ-induced surface PD-L1 is regulated by JAK 1 and 2 [33], STAT1, and STAT3, and the downstream transcription factor IRF1 [28]. Several cytokines, including IL-1α, IL-2, IL-7, IL-10, IL-21 IL-27, IL-32-γ, TNFα, and VEGF are inducers of PD-L1 expression on immune and tumor cells in the TME [5,34,35]. Proteins of several human endogenous retroviruses, including Epstein–Barr, hepatitis B, hepatitis C, human papillomavirus, and Merkel cell polyomavirus, are sources of inflamed tumors and high PD-L1 expression [36,37,38]. Chemotherapy and radiotherapy also influence PD-L1 expression in the TME [39,40]. Chemotherapeutics such as oxaliplatin, cyclophosphamide, doxorubicin, and paclitaxel have been shown to exert immunomodulatory functions through enhancement of antigen presentation or inhibition of immune suppressive mechanisms, thereby creating an inflammatory environment that can be invigorated by ICT [39]. Similarly, radiation induced cell death is followed by increased antigen presentation, IFN-γ response, and pro-inflammatory chemokine production that facilitate increased T cell trafficking to the tumors [41]. The immunostimulatory effects of radiotherapy lead to increased PD-L1 expression in the tumors that could synergize with ICT and develop as a systemic antitumor immunity [42,43].

PD-L1 is also regulated epigenetically at the mRNA levels by microRNAs (miRNAs). The observation that PD-L1 mRNA is readily detectable in many cell types but not at the protein level led to investigations on the role of miRNAs in regulating PD-L1 expression [29,44]. miRNAs are non-coding single stranded RNAs of 22–24 nucleotide length, which regulate gene expression either by inducing target mRNA degradation or by inhibiting translation. Nearly 20 miRNAs are known to regulate PD-L1 expression in several cancers, including NSCLC, pancreatic cancer, acute myeloid leukemia (AML), and colorectal cancer (CRC) [45]. Several miRNAs, including miR-513, miR-155, miR-570, miR-34a, and miR-200, have been described as suppressors of PD-L1 expression by direct binding to the 3′ UTR of PD-L1 mRNA [45]. Other miRNAs, including miR-20, miR-21, miR-130b, have been shown to repress PTEN causing increased PD-L1 expression [45].

The abundance and stability of PD-L1 expression on cancer cells are also stringently regulated by ubiquitination, N-glycosylation, and palmitoylation. Multiple studies have shown the role of cyclin D–CDK4 axis in the regulation of immune evasion mechanism by cancers and have established a link between cell cycle and immune evasion [46,47,48]. The cell cycle kinase CDK4 regulates PD-L1 expression via proteasome mediated degradation. Increased CDK4 expression is also associated with an inherent resistance program to anti PD-1 therapies that is driven by overexpression of IFN, Myc, and CDK4 signaling that leads to immune cell exclusion from the TME. When given alone, CDK4 inhibitors increase PD-L1 levels, reduce immune cell accumulation in tumors, and show synergism when given in combination with anti-PD-1 therapy. Furthermore, CDK4 inhibitors also enable the recruitment of NK cells to the tumors when given in combination with MEK inhibitor [48]. Another kinase, glycogen synthase kinase 3 beta (GSK3β), was shown to interact with PD-L1 and induce phosphorylation-dependent proteasomal degradation. That degradation can be antagonized by asparagine glycosylation (N192, N200, and N219) on PD-L1 [49]. In contrast, growth factors such as EGF are known to induce and stabilize PD-L1 in cells via GSK3β inactivation [49]. Supporting those in vitro observations, a strong correlation between phospho-EGFR and PD-L1 expression was observed, indicating that activation of EGFR contributes to an immunosuppressive TME by stabilizing PD-L1 expression [20,50,51]. Confirming those findings, EGFR inhibition by gefitinib leads to improved responses to antiPD-1 therapy by destabilizing PD-L1 [52]. Other cytokines, such as TNFα, that often play a major role in chronic inflammation also trigger immune escape from T-cell surveillance. However, while TNFα was shown to upregulate PD-L1 expression, TNFα induced miR155 was shown to suppress PD-L1 in human dermal lymphatic endothelial cells [53]. In cancer cells, PD-L1 stabilization is driven by TNF-1/p65/Cop9 signalosome 5 axis via deubiquitination activity of CSN5 [54]. CSN5 is an oncogenic protein that is essential for cell survival and plays a key role in adaptive immunity [55].

Protein pool and cell surface expression of PD-L1 are also regulated without changes in transcription levels. Two closely related members of the CKLF like MARVEL transmembrane domain containing proteins (CMTM4/6) contribute to increased PD-L1 protein levels [56,57]. CMTM6 co-localizes with PD-L1 on the plasma membrane, and binds and stabilizes PD-L1 at the cell-surface and in recycling endosomes, thereby protecting PD-L1 from lysosomal mediated degradation. CMTM6 also increases PD-L1 half-life by reducing ubiquitination [57]. Supporting the importance of surface stabilized PD-L1 levels in immune evasion, longer overall survival was seen in patients receiving ICT with high co-expression of CMTM6 and PD-L1 in stromal and immune cells (macrophages), but not tumor cells [57]. Another post-translational mechanism that regulates membranous expression of PD-L1 is lipid modification. Many of the protein functions involved with membrane association such as protein trafficking, activity, stability, and protein–protein interactions are controlled by palmitoylation, a reversible lipid modification process that involves the attachment of a 16-carbon fatty acid palmitate to proteins. In the case of PD-L1, palmitoylation occurs at Cys272 and blocks the ubiquitination required for PD-L1 degradation in the lysosomes [58]. As a consequence, palmitoylation increases the membranous half-life of PD-L1 and has been shown to promote breast tumor growth [59]. Blocking palmitoylation decreases interaction of PD-L1 with PD-1 and results in enhanced tumor immunity [58]. It is not known how those dynamic changes in PD-L1 expression influence tumor response to approved therapeutics. Thus, there is a pressing need to develop new approaches for non-invasive quantification of PD-L1 dynamics in real-time.

Similarly to PD-L1, PD-L2 is induced on tumor and immune cells upon exposure to IFN-γ. However, PD-L1 and PD-L2 are differentially regulated [28]. PD-L2 is mostly induced by IL-4 and GM-CSF in addition to IFNα and IFNβ with STAT3 and IRF1 transcription factors [28,60]. The differential regulation of PD-L1 and PD-L2 by cytokine signals indicates potential for molecular targeting and modulation of signaling pathways specific to either PD-L1 or PD-L2 in the context of cancer treatment, autoimmunity, or transplant tolerance. Detailed understanding of the signaling pathways used by cancer cells to counter the antitumor immune response could lead to development of improved prognostic markers that could be integrated with imaging studies.

## 3. PD-L1 IHC as a Biomarker for Guiding ICT

Molecular profiling and therapeutic selection centered on tumor-based biomarkers have transformed in delineating who should (or should not) receive a given therapy, resulting in an improvement of therapeutic outcomes in several cancers [65]. However, biomarker-based stratification has not been so straight forward for delivering ICT [66,67]. PD-L1 IHC is the most widely used biomarker for selecting patients to ICT [68,69]. Companion PD-L1 testing is approved for bladder, breast, cervical, gastric/gastroesophageal cancers, and NSCLC, and has proven useful in predicting the response in NSCLC with promising data emerging in bladder and gastric/gastroesophageal cancers [70,71,72,73,74,75,76,77,78,79,80,81,82,83,84,85]. Across those approvals, PD-L1 IHC is predictive of response in ≈29% of cases and not predictive in nearly 53% of cases [86]. Currently, it is unclear whether this is mostly due to artifacts related to IHC and tissue sampling or under-appreciated PD-L1 biology.

To define a PD-L1 positive or negative TME, tumors are broadly categorized into four types based on PD-L1 expression on the tumor cells and the presence of tumor infiltrating lymphocytes [87]: (1) PD-L1 positive and T cell positive; (2) PD-L1 negative and T cell negative; (3) PD-L1 positive and T cell negative; (4) PD-L1 negative and T cell positive. Earlier work in the evaluation of antiPD-1 therapeutics has focused on using PD-L1 expression as a biomarker to guide patient stratification based on the notion that those therapeutics block the interaction between immune cell PD-1 and tumor/immune cell PD-L1 [75,88,89]. Thus, the percentage of PD-L1 positive tumor or immune cells detected in the TME by IHC is termed as PD-L1 tumor proportion score (TPS) or immune proportion score (IPS), respectively [86]. Early clinical evaluation of nivolumab in solid tumors showed an improved overall survival in patients with high intratumoral PD-L1 expression [90]. In the KEYNOTE-001 clinical trial, TPS was the most robust biomarker associated with long-term survival with ≥50%TPS patients showing the most 5-year overall survival (OS) (29.6%) in advanced NSCLC patients treated with single-agent antiPD-1 therapeutic pembrolizumab [75]. Those findings led to FDA approval of PD-L1 detection by IHC as a companion diagnostic for receiving pembrolizumab [91]. Current evidence indicates that PD-L1 levels in the TME correlate with the magnitude of efficacy in NSCLC, although that association is not absolute [76,77,92,93]. In other cancers including Merkel cell carcinoma and triple negative breast cancer (TNBC), a correlation between response rate and IPS but not TPS was observed [86]. In few others, including cervical cancer, the combined positive score of PD-L1 (CPS) that accounts for TPS and IPS has been used to define PD-L1 positivity. That large variability in PD-L1 expression cutoffs, and the types of cells tested for PD-L1 expression, present significant challenges for PD-L1 IHC use in clinical practice [76,77,92,93].

The caveat of the PD-L1-based stratification is that not all the patients testing positive for PD-L1 respond. In addition to the technical aspects that are discussed in the following section, several biological factors also contribute to those differences. While PD-L1 levels in type 1 TME described earlier are driven by adaptive immune resistance, constitutive PD-L1 levels in type III tumors such as in NSCLC are driven by genetic alterations. Those PD-L1 positive tumors develop immune resistance to therapy through multiple mechanisms, including immunoediting, increased metabolism, and release of suppressive interluekins [94]. Some PD-L1 positive tumor types, as illustrated in the case of multiple myeloma (MM), also do not respond to ICT [95]. MM is a hematological malignancy of terminally differentiated plasma cells in the bone marrow [96]. PD-L1 is highly expressed in plasma cells of MM patients and associated with disease progression, as a further increase is observed in refractory and relapsed patients [95,97]. However, clinical evidence indicates that single agent anti-PD1 therapy is ineffective in MM [95]. Recent reports suggest that intrinsic heterogeneity of MM tumors, clonal expansion of tumor reactive PD-1^low^ T cells with a senescent phenotype, and expression of multiple immune checkpoint molecules in the TME could be some of the factors contributing to poor response [95].

Another caveat of PD-L1-based stratification is that some patients who tested negative for PD-L1 responded. Type II tumors often do not respond to PD(L)-1 therapeutics, as they are devoid of tumor intrinsic PD-L1 expression and adaptive immune resistance due to lack of T cells. However, it is conceivable that a PD-L1 negative state can be reversed by combination therapies that direct T cell accumulation in the tumors. Additionally, tumors with high mutational burden (TMB), virus driven cancers such as Merkel cell carcinoma (MCC), and tumor types with increased abundances of neoantigens—in the case of RCC, are highly immunogenic and known to respond to ICT in spite of PD-L1 negative status [80,98,99,100]. In some PD-L1 negative tumors, PD-L1 is induced on NK cells by the tumors via AKT signaling that can be activated by PD-L1 mAbs to control tumor growth [101]. Collectively, preclinical and clinical data suggest that understanding the relevance and regulation of PD-L1 in a particular tumor type is important for enhancing the utility of IHC and imaging assays.

Emerging data show that two highly used biomarkers, TMB and PD-L1, have non-overlapping effects on response to PD(L)-1 therapeutics and can be used as independent biomarkers for guiding ICT [83]. However, more recent studies in bladder cancer indicate an interplay between TMB and tumor PD-L1 levels, suggesting a need for deeper understanding of relationship between TMB and tumor PD-L1 levels. In the PURE-01 clinical trial in bladder cancer patients, a pathologic complete response in tumors with high TMB was closely associated with higher CPS, but no such association between response and CPS was observed in tumors with low TMB. Those observations lead to the development of a composite biomarker-based calculator that incorporates TMB and PD-L1 levels for guiding therapy [73]. Similar application-based approaches could enable the integration of imaging data for ICT response assessment.

In spite of clinical utility, several limitations associated with early PD-L1 diagnostic assays have contributed to challenges in equivalent interpretation of data across cancers and therapeutics [102]. Four IHC assays have now been approved by the FDA as a companion and complementary diagnostic for delivering PD-1 and PD-L1 therapeutics, and a fifth IHC assay is under development [103]. The following issues have to some extent compromised the accuracy and reliability of early IHC assays: Different mAb clones (22C3, 28-8, SP263, SP142, and 73-10), and different detection systems and technologies have been used for the assays, posing a challenge to the clinical application of these tests for therapeutic decision making. Three assays used TPS only, and the fourth assay used both TPS and IPS to stratify patients for therapy, thereby creating a scoring approach that is unique for each test and the drug. The pre-treatment or archival tumor tissue samples used to assess PD-L1 status may or may not have represented the tumor immune status at the start of the treatment. Small single biopsies or pleural effusions used for IHC assays represent a small fraction of the tumor, and are not reflective of the inter- and intra-tumoral heterogeneity. Alternative definitions of TPS cut off used for patient selection, ranging from ≥1% to ≥50%, inevitably lead to differences in clinical status classification and data interpretation [103,104,105]. To bring reproducibility in the application of PD-L1 tests, the blueprint project has performed analytical comparisons between the five mAbs used for IHC. Results of the project brought a consensus that 22C3, 28-8, and SP263 tests have sufficient analytical evidence for interchangeability, and the 73-10 assay has greater sensitivity than the other four tests [104,105]. However, many questions pertaining to the heterogeneity and PD-L1 dynamics remain unanswered and would benefit from the availability of non-invasive imaging.

It is not uncommon to observe heterogeneity in PD-L1 levels, both within and across lesions collected over time and/or from different anatomical sites in a single patient [106,107,108]. The observation that degree of PD-L1 positivity in one lesion does not predict the degree of positivity in another lesion, as in the case of NSCLCs, has significant therapeutic implications [107,109], and underscores the need for new methods for quantifying PD-L1 levels. Additionally, the location of the lesion is an important component associated with the immune response or lack thereof [110]. In urothelial cancers, NSCLCs, and melanomas, tumors that metastasize to the liver often result in poor prognosis and those that metastasize to lymph nodes demonstrate improved response rates [92,111,112]. Those results suggest that deeper understanding of tissue-specific immunoregulation is essential to developing effective immunotherapies tailored to tissue microenvironment [113]. Availability of highly specific high image contrast affording PET imaging agents that can quantify PD-L1 levels in the majority of the tissues and integrated into clinical workflow could assist in those endeavors.

Biopsy samples are often limited and needed for molecular tests that may confer sensitivity or resistance to other therapies. These issues are further compounded in cancers such as NSCLC and pancreatic cancers, which are difficult to obtain biopsies for and have patients with advanced stages of disease. For example, NSCLC tissue samples are needed for molecular tests for *EGFR*, *ALK*, and DNA repair genes that may confer sensitivity or resistance to other therapies, or for assessing tumor mutational burden (TMB) [114]. Furthermore, treatment options are impacted by PD-L1 expression levels in the first-line setting for NSCLC patients lacking targetable genomic alterations [114]. For those patients with multiple treatment options, treatment sequencing may be prioritized based on PD-L1 expression quantified by PET. For instance, patients with PD-L1 positive tumors might be advised to receive antiPD(L)-1 as first-line therapy, whereas patients with PD-L1 negative tumors could receive it as second-or later-line therapy [115,116,117,118]. Access to PD-L1 PET would enable a multiple biomarker strategy and better use of biopsies.

## 4. PD-L1 Imaging Agents

### 4.1. Monoclonal Antibodies and Small Proteins

When suitable radiotracers are available, use of nuclear imaging techniques, such as PET and single photon emission computed tomography (SPECT), is an advantageous strategy for non-invasive quantification of proteins and biochemical processes in vivo. PET has been found very effective for repeated measurements of target expression in tumors [119], predicting responses to therapy and progression-free survival, and aiding in drug development and evaluation [120,121]. To address the need to quantify PD-L1 levels non-invasively, an array of imaging agents, including mAbs [122,123,124,125,126,127,128,129], mAb fragments [130,131,132], a small protein [133], and peptides [134,135], have been developed and investigated in preclinical models. A few of those agents are in early clinical investigations and have shown promising results.

The majority of the reported studies have taken advantage of the high specificity, affinity, and ready availability of mAbs targeting PD-L1. The accessible ε-amino groups of lysine residues of mAbs are amenable to modification with various bifunctional chelators. Those mAb conjugates are then chelated with radiometals with long halve-lives that match the pharmacokinetics of antibodies. ^89^Zr (t_1/2_, 3.27 d) and ^111^In (t_1/2_, 2.8 d) are two of the most commonly used radionuclides for PET and SPECT imaging respectively. Another radionuclide, ^64^Cu (t_1/2_, 12.7 h) is also becoming a radiolabel of choice for antibodies [136]. The challenge posed by the short half-life of ^64^Cu is to some extent offset by the sensitivity and resolution afforded by PET. Direct radiolabeling of antibodies is also accomplished using radiohalogenation of tyrosine with PET radionuclide I-124 or SPECT radionuclides I-123/125.

A diverse selection of mouse (mPD-L1) [126,127,137] or human PD-L1 reactive (hPD-L1) [122,123,128,129,136,138,139,140,141,142] mAbs have been investigated for their potential to detect PD-L1 levels in vivo. Most of those studies were conducted in immune compromised mice with human tumor xenografts, or in immunocompetent mice with syngeneic tumors. Studies by Heskamp et al. demonstrated the feasibility of imaging PD-L1 levels in vivo using an anti-hPD-L1 mAb labeled with ^111^In [129]. SPECT imaging and biodistribution studies showed high uptake and retention of [^111^In]mAb in human breast cancer xenografts with high PD-L1 expression but not in tumors with low PD-L1 expression. In early evaluations of a target or imaging agent experiments, using knockout mouse models provides a unique advantage for delineating the target uptake specificity. Evaluation of ^64^Cu-labeled anti-mPD-L1 mAb in wild type and PD-L1 deficient mice offered insights into the PD-L1 levels in the whole body. High uptake of radioactivity seen in lungs of WT type but not in PD-L1 deficient mice could perhaps explain the lung related toxicities associated with ICT and the role PD(L)-1 axis plays in restraining immune responses in the lung [127]. A significant reduction in uptake was also seen in secondary lymphoid organs (spleen and lymph nodes), known for physiological expression of PD-L1, in PD-L1 deficient mice [127]. Those observations indicate the opportunities for the use of PET to evaluate changes PD-L1 levels in tissues that play an essential role in immune priming.

Almost all newly developed therapeutic mAbs are humanized or fully human mAbs. Evaluations of radiolabeled versions of these agents in vivo using imaging provides unique insights into the PK and PD of the corresponding therapeutic agents. To characterize those PK/PD changes, we investigated [^111^In] labeled atezolizumab ([^111^In]atezolizumab) [128]. Atezolizumab is a PD-L1 mAb that is approved to treat NSCLC, RCC, and Head and neck squamous cell carcinoma (HNSCC). [^111^In]atezolizumab uptake in multiple human tumor xenografts reflected the graded levels of PD-L1 expression characterized ex vivo by IHC. In those studies, increased protein dose resulted in increased accumulation of [^111^In]atezolizumab in the tumors and reduced uptake in non-specific tissues [128]. Although low doses of mAbs are often used in imaging studies, dose-dependent tumor accumulation is not uncommon with mAbs, particularly when the target is internalized. At a low protein dose, low target binding and accumulation can be observed in the tumor due to non-target-specific and nontumor but target-specific uptake of mAb in other tissues that could drastically reduce serum mAb concentration and target-specific tumor accumulation. This phenomenon is termed targeted-mediated drug disposition (TMDD) [143,144]. Nonspecific binding sites are saturated when higher mAb doses are used, resulting in higher systemic exposure and increased binding to the target in the tumors. Studies with atezolizumab chimera have also shown that increased mAb dose could facilitate deeper penetration of the mAb into the tumors [138]. These studies support identifying an optimal protein dose required to maintain mAb pharmacokinetics to achieve high contrast images.

Immune system targets pose unique challenges for in vivo evaluation. Cancer research relies on preclinical models for drug development; however, immune biology of human cancers is not always well reflected in the routinely used syngeneic mouse tumor models. Therapeutics developed to bind human PD-L1 for clinical translation do not always possess cross-reactivity with murine PD-L1. That lack of cross reactivity constrains drug candidate evaluation in immunocompetent mouse models to assess immune system induced changes in the tumor. Thus, mAbs with cross reactivity to both mouse and human orthologs can provide unique insights into mAb distribution in vivo and a bridge between preclinical and clinical studies and observations. The advantage of using such agents with human and mouse cross reactivity was reflected in radiolabeled atezolizumab and avelumab biodistribution in vivo [122,125,128,145]. Atezolizumab and avelumab bind both human and mouse PD-L1, although with differences in affinity. Increased accumulation of radiolabeled atezolizumab was observed in brown adipose tissue (BAT) in mouse models [128], a novel finding that has also been observed with anti-mouse PD-L1 mAbs but not with anti-hPD-L1 mAbs [127,128,129].

mAbs have proven to be a highly reliable resource for imaging agent development. However, their large sizes and longer half-lives require several days of washout to generate high contrast images. Hence, there is a significant interest in developing imaging agents with faster PKs that would provide high contrast images within few hours post-injection [146]. Several mAb fragments, including single domain antibodies (sdAb) [147,148], nanobodies [149], small proteins derived from adnectin [133], and PD-1 external domains [150], have been developed and investigated for their potential to detect graded levels of PD-L1 expression [150]. Most of those agents have demonstrated faster clearance and provided high contrast images within several hours of administration. For example, high contrast PD-L1 specific images can be obtained with the adnectin-derived PD-L1 imaging agent, ^18^F-BMS-986192, within 120 min of injection [133]. Notably, observations made using mAbs are also reproduced using these small proteins. PD-L1 PET imaging is also finding new uses in other diseases, like metabolic disorders. BAT is involved in metabolic and temperature maintenance, and the metabolic activity of BAT is often measured using ^18^F-FDG, a radiotracer for glucose metabolism. Studies using radiolabeled vHH domains have established that PD-L1 is expressed by BAT cells in murine models independent of metabolic state of BAT, providing a new way for non-invasive quantification of BAT activity [132].

### 4.2. Peptides and Low Molecular Weight Agents

mAbs and mAb fragments are less efficient than low molecular weight agents (LMWs), such as small-molecules and peptides for tissue penetration, tumor retention, and blood clearance, owing to their large size [151]. Therefore, our group has focused on LMWs that often have faster PK. High affinity peptide inhibitors of PD-L1 (<10 nM) were reported only within the last few years [152]. We selected a peptide, WL12 (Figure 1A), that is most suitable for conjugation and possessed a single primary amine from the reported peptide library [153]. Next, we assessed its binding mode to PD-L1 (PDB ID: 4ZQK) [154] by docking studies, wherein we superimposed WL12 with PD-1. Our model showed that WL12 forms a beta sheet-like structure, and binds in a similar mode to that of PD-1 (Figure 1B). Importantly, the primary amine of the ^13^Orn residue is exposed and does not participate in binding with PD-L1. That finding led us to develop a ^64^Cu-labeled DOTAGA conjugated peptide, [^64^Cu]WL12, which demonstrated high affinity (2.9 nM), and in vitro and in vivo specificity to PD-L1 in Chinese hamster ovary cell line stably expressing PD-L1 (hPD-L1) and in TNBC MDAMB231 cells with high PD-L1 expression, compared to controls (CHO and SUM149). [^64^Cu]WL12 also demonstrated specific uptake in PD-L1 positive tumors (Figure 1C,D), high tumor-to-muscle and tumor-to-blood ratios by 60 min, and the PD-L1 specific binding could be blocked by saturable concentrations of cold peptide [153]. Importantly, the high specificity and high contrast images provided by [^64^Cu]WL12-PET also enabled the quantification of dynamic changes in PD-L1 levels in the tumors, indicating potential to quantify therapy induced changes in PD-L1 levels. LMWs also offer synthetic tractability and are also easily amenable to structural changes to modulate the PK to meet the needs of imaging modality and application. WL12 labeled with ^68^Ga demonstrated increased clearance from non-specific tissues and provided images with improved contrast than the ^64^Cu-analog [135]. In contrast, replacing DOTAGA with ^18^F-fluoro nicotinic acid for radiolabeling with ^18^F reduced the hydrophilicity of the molecule. Although the [^18^F]WL12 showed PD-L1 specific uptake in tumors, the tumor-to-background ratios were low and higher radioactivity uptake was seen in non-specific tissues [134]. These studies exemplified the tractability of LMWs for modification to affect in vivo distribution to optimize the image contrast.

### 4.3. PD-L1 PET in Patients

Quantification of PD-L1 levels in patients was reported in three first-in-human studies using three distinct molecular entities (mAb, adnectin, and sdAb) [155,156,157] labeled with three different radionuclides (^89^Zr, ^18^F, and ^99m^Tc). All three agents were found to be safe and effective and showed promise in detecting PD-L1 levels in vivo. In spite of the distinct nature of the imaging agents, a few common features in distribution among the radiotracers can be derived. High uptake in tumors, metastases, and secondary lymphoid organs, and heterogeneity in tumor uptake between and within patients were observed with all the three agents. Distribution in other tissues was different between radiotracers due to the nature of metabolism and clearance of different molecular entities. Collectively, these studies indicate that PD-L1 PET imaging can provide insights into PD-L1 levels and dynamics to guide clinical management for patients receiving ICT.

The feasibility of quantifying PD-L1 levels in the whole body by PET was demonstrated using ^89^Zr-labeled atezolizumab (Figure 2). Investigations of [^89^Zr]atezolizumab in patients with NSCLC, TNBC, and bladder cancers have captured the PD-L1 levels in all the lesions in entirety and the inter and intra-tumoral heterogeneity in PD-L1 expression. Differences in uptake were also observed between different cancer types, with bladder cancers showing the highest uptake of [^89^Zr]atezolizumab followed by NSCLC and TNBC. [^89^Zr]atezolizumab uptake was related to size change in target lesion and higher in patients who showed better responses, warranting further investigation. In the small number of patients investigated, it was not PD-L1 IHC or RNA-sequence-based predictive biomarkers but baseline standard uptake values (SUVmax) of [^89^Zr]atezolizumab that indicated a better outcome for patients. High [^89^Zr]atezolizumab uptake was observed in lymph nodes and the spleen, reflecting the physiological expression of PD-L1. High [^89^Zr]atezolizumab uptake in sites of inflammation was also noted, indicating the possibility for false positives and the necessity for cross-correlation of imaging findings with IHC. [^89^Zr]atezolizumab uptake was high in the liver, intestines, and kidneys, possibly reflecting the metabolism and clearance of mAbs.

Early studies with another PD-L1 imaging agent, ^18^F-BMS-986192, have also shown promising results in NSCLC patients [156] (Figure 3). ^18^F-BMS-986192 uptake in the NSCLC lesions was heterogeneous within and between patients and the median uptake (SUVpeak) was higher in patients with >50% TPS compared to patients <50% TPS. ^18^F-BMS-986192 uptake was also observed in spleen and lymphoid tissues, similarly to the other preclinical and clinical studies. In addition, non-specific uptake was observed in liver due to catabolism, and in kidneys and GI tract due renal and biliary clearance. The small size of ^18^F-BMS-986192 also facilitated image acquisition within 60–90 min.

An sdAb with a histidine tag, NM-01, was labeled with SPECT radionuclide ^99m^Tc and investigated in NSCLC patients for safety, imaging characteristics, and dosimetry [157]. ^99m^Tc-NM-01 uptake in NSCLC lesions could be clearly seen in SPECT images and showed heterogeneity in uptake within and between patients. A modest correlation was observed between PD-L1 immunoreactivity by IHC in biopsies and ^99m^Tc-NM-01 uptake measured as tumor-to-blood pool activity ratio. While ^99m^Tc is advantageous due to its long half-life and ready availability, the low spatial resolution (1.8 mm for PET vs. 10 mm for SPECT) and lack of quantitation with SPECT could pose challenges in quantifying the PD-L1 levels in small lesions and therapy induced PD-L1 dynamics.

## 5. PD-L1 PET for Non-Invasive Quantification of PD-L1 Dynamics

Beyond detecting PD-L1 expression in malignant and normal tissues, mAb-based PD-L1 PET has also been investigated to detect PD-L1 dynamics induced by chemotherapy and radiotherapy. Significantly higher uptake of ^89^Zr-radiolabed mAbs in human tumor xenografts reflected paclitaxel-induced PD-L1 expression [123]. Radiotherapy is also known to induce PD-L1 expression in the TME, primarily due to the increased accumulation of immune cells in response to radiation-induced inflammation. However, enhanced PD-L1 expression is not observed in every tumor. Thus, identifying those tumors with enhanced immune response to radiotherapy could improve the potential to harness those responses for tumor control by ICT. Increased PD-L1 expression was observed in tumors of mice receiving fractionated radiation doses. Studies using [^89^Zr]atezolizumab and other anti-mouse PD-L1 mAbs have demonstrated enhanced uptake in irradiated human tumor xenografts in immunocompromised mice and in syngeneic B16F10 melanoma tumors in immunocompetent mice [124,145]. These studies demonstrated the potential of radiolabeled mAbs to quantify radiotherapy induced changes for therapy guidance.

The rapid clearance and high sensitivity of small proteins have also allowed for quantification of molecular therapy induced changes in PD-L1 levels in tumors. MEK inhibitor treatment leads to downregulation of PD-L1 expression in tumors that could potentially be quantified using PET. Studies with [^18^F]BMS-986192 have demonstrated a significant reduction in imaging agent uptake reflecting the 50% down regulation in PD-L1 levels following treatment with MEK inhibitor selumetinib for 24 h [158]. Additionally, EGFR inhibitor induced changes in PD-L1 downregulation were quantified using ^89^Zr-Df-KN035, a nanobody derived radiotracer [159]. These preclinical observations have the potential to be validated in a clinical setting as the therapeutics used have been approved or are undergoing clinical investigation.

## 6. PD-L1 PET for Dose Optimization and Drug Development

To date, all approved PD-L1 drugs have been mAbs. Atezolizumab (non-glycosylated IgG1κ, t_1/2_ 27 d) is approved for liver cancer [160], metastatic NSCLC [2], extensive stage small-cell lung cancer, metastatic TNBC [161] and advanced urothelial cancers [162]. Avelumab (IgG1λ, t_1/2_ 6.1 d) is approved to treat metastatic Merkel-cell carcinoma [163] and advanced or metastatic urothelial carcinoma [164]. Durvalumab (IgG1κ, t_1/2_ 17 d) is used to treat advanced or metastatic urothelial cancers [165] and unresectable stage-III NSCLC [166]. Concentrations and target engagement potential of these therapeutics at the tumor remain unknown and are influenced by PD-L1 levels. This is an area of investigation that could benefit from non-invasive technologies and better understanding of PD-L1 regulation. The therapeutic effect of those drugs is achieved through direct binding of the drug to the target. This binding often occurs at a functional site, and for PD-L1 therapeutics this is where PD-1 binds. Thus, efficacy of a PD-L1 mAb is critically dependent on the extent of its target engagement.

Unlike small molecule drugs, interactions of PD-L1 mAbs with the target often affect the PK of the mAb [143]. TMDD is common for mAb therapeutics directed against proteins expressed on cell membranes and receptor-mediated endocytosis results in drug elimination. PD-L1 internalization presents an important clearance pathway for PD-L1 antibodies. Therefore, modulation of PD-L1 by mAbs, other regulators, concomitant treatments, or disease progression can result in changes in the PK, and subsequently, the PD effects of the mAb. Occupancy data for PD-L1 therapeutics, reported in only a limited number of trials, are obtained using peripheral blood mononuclear cells (PBMCs) [167], and occupancy at the tumor is almost always predicted by in silico modeling. The technologies for real-time assessments of drug-target engagement in vivo are too few to incorporate real-time data into those models [168,169]. Often, an mAb disposition in vivo is obtained in preclinical and clinical studies using radiolabeled antibodies, such as atezolizumab. A limitation of using radiolabeled mAbs is that observations made are highly specific to the mAb under investigation, and depend on mAb properties such as valency, shape, size, isoelectric point, and dosage, each of which influences the mAb’s PK [155,170,171]. Additionally, radiolabeled-mAbs provide a cumulative measure of exposure at the tumor and target expression, and have low sensitivity to capture therapy induced changes in PD-L1 levels [1,172]. This is further complicated by the thousands of combination therapies that are being investigated with few technologies available to determine drug concentrations at the tumor and to evaluate their relevance to therapeutic outcomes [9].

In molecular modeling studies, we observed that WL12 interaction surface on PD-L1 overlapped with that of PD-1 and PD-L1 mAb interaction surfaces on PD-L1 [173] (Figure 1B). The low affinity of WL12 compared to antiPD-L1 mAbs (>2 nM vs. <2 nM) led us to postulate that peptide-based PET could be used to quantify accessible PD-L1 levels during treatment with a PD-L1 mAb. Such quantitative measurements can inform the target engagement potential of any mAb at the tumor. To test this, several PD-L1 expressing cancer cell lines were incubated with [^64^Cu]WL12 with and without antiPD-L1 mAbs. The bound radioactivity measurements showed significantly less cell bound [^64^Cu]WL12 in the presence of mAbs indicating fewer accessible PD-L1 levels [173]. We then tested if those observations could be reproduced in vivo. In NSCLC and TNBC human tumor xenograft-bearing mice treated with a saturating dose of all anti-PD-L1 mAbs (atezolizumab, avelumab, and durvalumab), a significantly reduced [^64^Cu]WL12 uptake was observed compared to saline treated controls (Figure 4). These results established that target engagement of any anti-PD-L1 mAb that inhibits PD-L1:PD-1 interaction can be quantified using [^64^Cu]WL12. Furthermore, these studies laid the foundations for defining the effect of mAb dose on target engagement in the tumor to define the dose required for saturating PD-L1 levels in the tumor [173]. Such measurements have the potential to guide future drug dosing and optimization and could become critical in identifying the dosage for novel class of mAb therapeutics such as probodies [174] and bispecific mAbs.

## 7. Future Considerations

Impressive clinical results seen over the last decade clearly established the role of the immune checkpoint blockade in restoring anti-tumor immunity. The durable and dynamic immunity observed does not benefit all the patients receiving those therapies. To further enhance those therapeutic outcomes, there is a growing interest in identifying on-treatment and post-treatment biomarkers, applicable to a variety of cancers, that could guide therapy. In addition to PD-L1 IHC, several other leads, including TMB, aberrant oncogenic signaling, and immune regulatory molecules, are also being investigated. PD-L1 IHC is arguably the most validated biomarker that is amenable for imaging agent development. Evaluation of PD-L1 levels by PD-L1-PET imaging is expected to provide a better understanding of response to ICT; however, several challenges need to be addressed. Specifically:

Imaging agents with better PK: All three agents in early clinical evaluations are proteins. In spite of the promising results seen, these molecular entities generally produce inferior contrast compared to peptides or small molecules. Additionally, the distribution of the evaluated imaging agents indicates non-specific uptake in liver, kidneys, and intestines that could further limit their applicability for malignancies originated in those tissues. Considering that PD-L1 density is a few hundred thousand ligands/cell, imaging agents with improved contrast and high target-to-background ratios are essential for successful and broader application of PD-L1 PET to guide ICT.

Specific activity of imaging agents: High specific activity is crucial for quantifying low density targets such as PD-L1. All the agents tested in clinical trials are biologicals and require a high protein dose to achieve optimal image contrast. The sizes of the molecules (>10 kDa size) also make it difficult to achieve separation of radiolabeled agents from non-radiolabeled precursors. These factors result in an effective specific activity of few hundred mCi/μmole for these agents. Such low specific activities could limit our ability to quantify changes in PD-L1 levels below a certain threshold and needs to be investigated. Those limitations can be addressed by peptides or small molecule-based agents that can be synthesized with high specific activities of several thousand mCi/μmole.

*Time of PD-L1 PET sampling*: Immune responses to ICT are unpredictable and vary by the type of tumor and individual immune fitness. Thus, the optimal time for imaging during on-treatment and post-treatment may need to be determined for each tumor type and may have an impact on defining PD-L1 signal cut-off for positivity.

PD-L1 expression as a continuous variable: It is becoming evident that differences in outcomes in some of the prior studies may be the result of imbalances in PD-L1 distribution during patient randomization rather than the treatment intervention [117]. A retrospective analysis of NSCLC clinical trial data indicated that PD-L1 levels need to be interpreted as a continuous variable instead of the a priori cut points now used to define PD-L1 high vs. low populations. Those lessons need to be incorporated into PET imaging measures for making treatment decisions.

Correlation of PET measures with CPS: In recent years, there has been a shift towards using CPS to correlate with response. PET measurements provide total PD-L1 levels in the tumors that cannot be separated into the contributions from TPS and IPS. CPS has shown promise in some cancers, including bladder and gastroesophageal cancers, although its relevance needs to be established for every tumor type. Total PD-L1 levels measured as CPS are more relevant than TPS and IPS to correlate with PET measures.

Integration of PET with genomic data: As discussed in the previous sections, PD-L1 is regulated at multiple levels and their contribution is difficult to discern. Thus, integration of PET data with serum or tissue-based biomarkers could assist in delineating the underlying biology and improve the diagnostic accuracy of PET measures.

Uptake in autoimmune and infectious diseases: The PD(L)-1 axis is central to regulating the auto-reactive T-cell responses and mediates tissue tolerance and immune-cell mediated damage during viral infections [175]. Elevated PD-L1 levels were observed on disease tissue cells in several autoimmune diseases, including IBD, Sjögren’s syndrome, and on activated T cells of rheumatoid arthritis patients [176,177,178]. Elevated PD-L1 levels were observed in the livers of patients with hepatitis C infections and in the splenic cells of patients with LCMV infections [178]. The importance of PD (L)-1 in dysregulated immune homeostasis and increased accumulation of [^89^Zr]atezolizumab in inflammatory lesions underscore that patterns of uptake in tissues not directly related to patients cancer need to be carefully considered during interpretation.

## 8. Conclusions

Recent first-in-human evaluations of PD-L1 imaging agents have brought molecularly targeted, non-invasive quantitative measures to guide and assess the immune response to ICT. Many challenges pertaining to interpretation and integration of PET data still need to be resolved. It is becoming clear that there is no “one biomarker fits all” approach for monitoring immunotherapy. Thus, a composite application/score that incorporates multiple biomarkers, including imaging data, will be highly valuable. Such novel approaches that incorporate adaptive components present burgeoning opportunities for timely treatment planning to achieve durable responses in cancer patients.

## Figures and Tables

**Figure 1 cancers-12-03173-f001:**
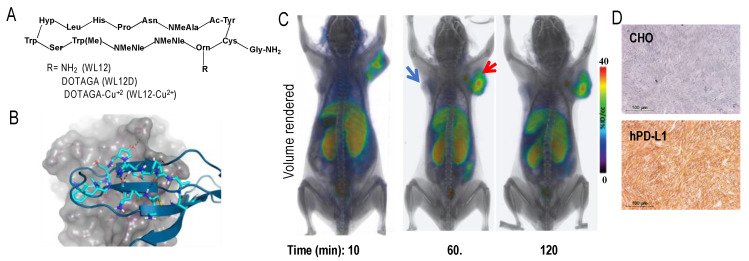
WL12 binding interactions with PD-L1 overlap with those of PD-1. (**A**) Structural representation of WL12 and its analogs; (**B**) predicted binding mode of WL12 to PD-L1. WL12 forms a beta sheet-like structure in the groove of PD-L1. WL12 is shown in cyan. The surface representation of PD-L1 is shown in gray, with the ribbons and key side chains shown in magenta; WL12 mimics PD-1 binding to PD-L1. The structure of PD-1 is shown in teal. The two main interacting beta strands of PD-1 overlap well with the conformation adopted by WL12 bound to PD-L1. (**C**) NSG mice with hPD-L1 (red arrow) and CHO tumors (blue arrow) were administered intravenously with 150 μCi of [^64^Cu]WL12 and images were acquired at 10, 60, and 120 min after the injection of the radiotracer. 3D volume rendered images show specific accumulation of [^64^Cu]WL12 in hPD-L1 tumors. (**D**) PD-L1 IHC shows strong immunoreactivity (brown color) in hPD-L1 tumors (from [153]).

**Figure 2 cancers-12-03173-f002:**
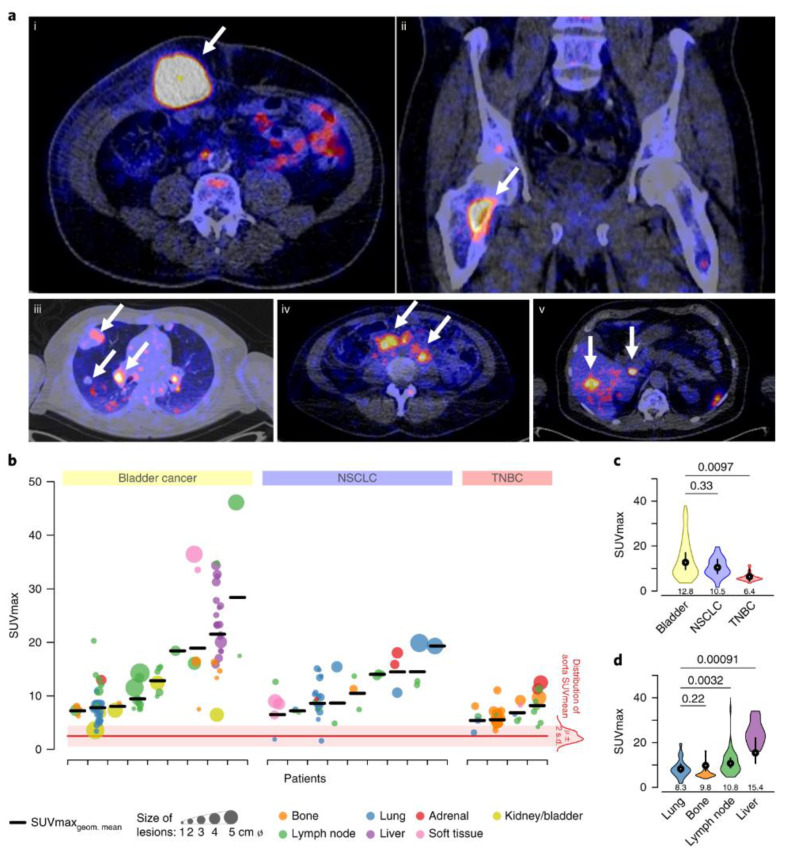
^89^Zr-atezolizumab tumor uptake. (**a**) Examples of PET/CT images of four patients illustrating ^89^Zr-atezolizumab tumor uptake in five different locations on day 7 post injection (white arrows indicate tumor lesions; PET scans were performed once per patient and time point). Images (i) and (ii) are from the same patient, whereas images (iii), (iv), and (v) are from a separate patients. (**b**) Overview of ^89^Zr-atezolizumab uptake as SUVmax at day 7 post injection in 196 tumor lesions with a diameter >2 cm grouped per tumor type and ordered by increasing geometric mean SUVmax per patient, visualizing tumor size and site, and with blood pool background uptake as reference. Horizontal bar indicates geometric mean SUVmax per patient. (**c**) Violin plot of actual distribution of SUVmax in lesions per site of lesion with bottom and top 1% of SUVmax values truncated (first, fiftieth, and ninety-ninth SUVmax percentiles: 1.7, 7.9, 19.6 for lung; 3.9, 5.6, 16.4 for bone; 4.6, 9.7, 40.1 for lymph node; 16.1, 23.3, 34.1 for liver); black vertical lines are 95% CIs of geometric mean SUVmax; white dots within black lines and values below the violin plot are the actual geometric means, all based on a linear mixed regression model with two-sided Wald *p* values using Satterthwaite approximations to degrees of freedom under restricted maximum likelihood, shown above the graph; *n*_lung_ = 44 in ten patients, *n*_bone_ = 62 in nine patients, *n*_lymph node_ = 54 in 20 patients, *n*_liver_ = 19 in one patient. (**d**) Violin plot of SUVmax in lesions per tumor type with bottom and top 1% of SUVmax values truncated (first, fiftieth, and ninety-ninth SUVmax percentiles: 3.6, 10.9, 38.0 for bladder; 1.7, 9.7, 19.6 for NSCLC; 3.4, 5.6, 11.7 for TNBC); black vertical lines are 95% CIs of geometric mean SUVmax; white dots within black lines and values below the violin plot are the actual geometric means, all based on a linear mixed regression model with two-sided Wald *p* values using Satterthwaite approximations to degrees of freedom under restricted maximum likelihood, shown above the graph; *n*_bladder_ = 85 in nine patients, *n*_NSCLC_ = 43 in nine patients, *n*_TNBC_ = 68 in four patients from [155].

**Figure 3 cancers-12-03173-f003:**
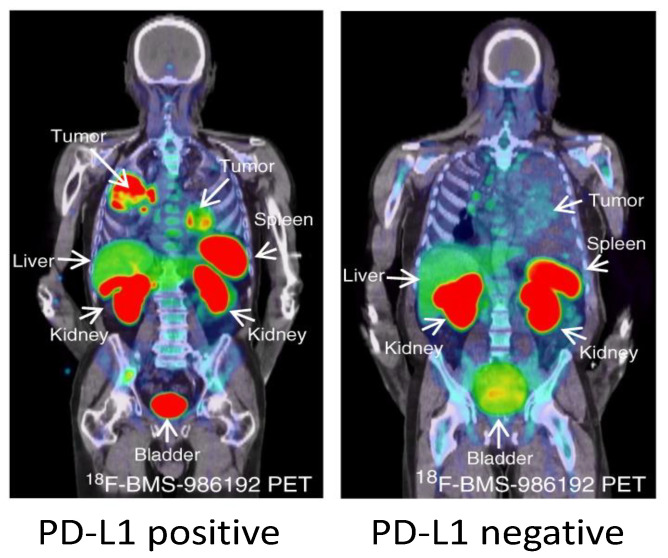
^18^F-BMS-986192 tumor uptake in patients. Patient 2 with tumor with PD-L1 expression > 95% (left panel). ^18^F-BMS-986192 PET (145.7 MBq, imaging time point 1 h post injection (p.i.)) demonstrates heterogeneous tracer uptake within and between tumors. Patient 3 with tumor PD-L1 expression < 1% (right panel). ^18^F-BMS-986192 PET (214.62 MBq, 1 h p.i.) demonstrates low tumor tracer uptake. Physiological expression=driven uptake in the spleen and non-specific uptake in the liver were observed with ^18^F-BMS-986192. Red and blue in the images represent maximum and minimum accumulated radioactivity, respectively, if a rainbow color scale is used(from [156]).

**Figure 4 cancers-12-03173-f004:**
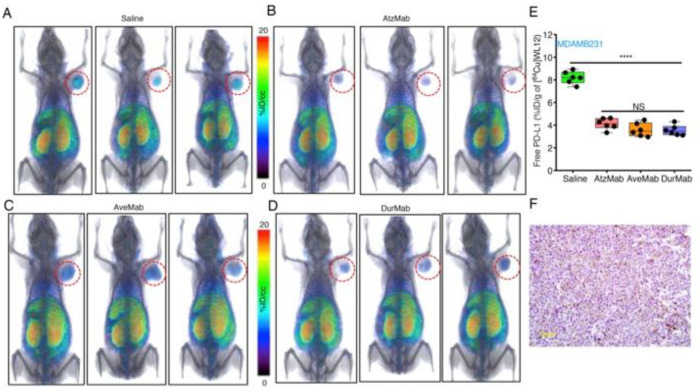
Tumor PD-L1 engagement by 3 distinct PD-L1 therapeutic antibodies quantified using [^64^Cu] WL12. (**A**–**E**) [^64^Cu]WL12 uptake in MDA-MB-231 xenografts is significantly reduced in mice receiving AtzMab (20 mg/kg), AveMab (10 mg/kg), or DurMab (10 mg/kg) 24 h prior to radiotracer injection. Whole-body volume-rendered [^64^Cu]WL12 PET-CT images of saline- (**A**) AtzMab- (**B**) AveMab- (**C**) and DurMab-treated (**D**) mice, and ex vivo biodistribution (**E**) at 2 h after [^64^Cu]WL12 injection (*n* = 6–9/group). (**F**) IHC staining for PD-L1 in the corresponding tumor. Scale bars: 100 μm. Box-and-whisker graphs showing minimum to maximum and all data points, with the horizontal line representing the median. **** *p* < 0.0001; NS, not significant, by 1-way ANOVA and Dunnett’s multiple comparisons test in **E** (From [173]).

**Table 1 cancers-12-03173-t001:** Regulators of PD-L1.

Type	Regulators of PD-L1	Change in PD-L1 Levels	References
Genetic alteration	amplifications translocations disruption of 3′ UTR region	↑	[14,15,16]
Oncogenic signaling	CDK5 STAT3 PTEN loss MEK RAS EGFR ALK MYC	↑	[18,20,21,24,61,62,63]
Inflammatory signaling	INF-γ INF-α IFN-β TNFα TGF-β	↑	[5,28,29,54,64]
Post-translational modifications	CMTM6 CMTM4 CSN5 palmitoylation	↑	[48,49,54,56,57,58]
	GSK3-β CDK4	↓	

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
