# Peer review of "Quantifying PD-L1 Expression to Monitor Immune Checkpoint Therapy: Opportunities and Challenges"

_cancers, 2020, doi:10.3390/cancers12113173_

Round 1
Reviewer 1 Report
The review is now improved and can be accepted.
Reviewer 2 Report
Very nice revision and a comprehensive review on the role and detection of PD-L1.
This manuscript is a resubmission of an earlier submission. The following is a list of the peer review reports and author responses from that submission.
Round 1
Reviewer 1 Report
Authorship: ...and PhD. What does it mean?
Abstract: poorly written, with short sentences that are sometimes ripetitive. It should be improved and made more reader-friendly.
Introduction: Why only anti-PD-L1 are mentioned? they should be discussed along with anti-PD-1, the biological and clinical scenario is the same.
A meta-analysis in JAMA oncology by Duan et al. has recently indicated that anti-PD-1 antibodies might be clinically more effective than anti-PD-L1 antibodies. These data must be discussed in depth.
There is no adequate mention of other strategies for the selection of patients who might benefit from anti-PD-1 or anti-PD-L1 antibodies, such as evaluation of T cell clones, IL-8 measurement, investigation of B cell tertiary structures, enumeration of lymphoid and myeloid subpopulations in the peripheral blood, PD1 mRNA enumeration, mutational burden.
Reviewer 2 Report
This is a well-written review examining the potential and limitations of PDL1 quantitation for immune checkpoint inhibitor patient selection and response monitoring. I have mostly minor comments listed below, although there was insufficient discussion on the actual value of PDL1 quantitation. Many of the limitations were mentioned, including the dynamic expression and inter -and intra-tumoural heterogeneity of expression. In addition to this, however, many patients with PD-L1 negative tumours will respond to immune checkpoint inhibitors – so the pertinent question of whether PD-L1 status should be used to select patients for these therapies should be included and discussed in greater detail. Minor comments Introduction: line 5 from end of page: Should read ‘Although PDL2 possesses high affinity’ Page 2, last sentence paragraph 1 should read ‘The majority of PD1 activity Page 3, The comment that CDK4 promoted PDL1 degradation and CDK4 inhibition improves OS in combination with anti PD1 needs some additional details, why would CDK4i induced PDL1 stabilisation be beneficial in PD1 inhibitor therapy Page 3 The sentences on EGFR are unclear – EGF can stabilise PDL1, but patients with high p-EGFR also have low PD-L1? Why the difference in these studies? Page 3 The role of TNF-1 in PD-L1 induction is much more complex, with many reports indicating that TNFa does not induce PD-L1 expression There is inconsistency in PDL1 and PDL2 labelling - i.e use either with or without hyphen for both and throughout the manuscript Page 4, second last paragraph: The sentences itemising some of the limitations of the IHC PDL1 assays are not well written Page 4 The blueprint PD-L1 assay needs referencing and some details of the outcome of this project should be included Page 5, paragraph 1: The section on PET measurement of PD-L1 is a little repetitive Figure quality is poor in this pdf document Explanation of the RGB colour scale should be provided in Figure 3, and legend should also highlight the non-specific uptakeAuthor Response
Please see the attachment

Reviewer 3 Report
Nimmagadda reviewed the regulation of PD-L1 expression, PD-L1 imaging agents, PD-L1 PET, and challenges these advances present in the selection of immune checkpoint therapy (ICT) and its schedule. The manuscript is well written and and useful for a better understanding of PD-L1 regulation and the use of ICT, although the following points should be addressed.
Comments:
In the section on PD-L1 regulation:
- miRNA and palmitoylation play crucial roles in the regulation of PD-1/PD-L1 expression. Thus, it would be preferable to explain the mechanisms of PD-L1 regulation.
- The author mentioned different signaling pathways involved in PD-L1 upregulation in several kinds of cancers. Other cancers including myeloma, in which survival is supported by the tumor microenvironment, should also be included.
- The author noted that longer overall survival times were seen in patients with high PD-L1 expression in the tumor microenvironment who were treated with ICT. Another report showing the importance of PD-L1 expression in the tumor microenvironment should be mentioned in this review.
In the section on PD-L1 IHC:
- The author noted a positive correlation between PD-L1 expression and overall survival in patients with several types of tumor. It is important to specify which tumors are being referred to.
In the section on PD-L1 PET:
- There are numerous explanations of PD-L1 PET throughout this manuscript, which should be consolidated in the section on PD-L1.
- PD-L1 PET is a promising tool to measure total PD-L1 levels in tumors. Some cancer patients have focal infections that might be positive in PD-L1 PET. Is it difficult to distinguish infected sites from tumors? Further, it would be preferable to discuss metabolic PD-L1 volumes in tumors.
In the overall manuscript:
- A conclusion section should be included.
- There are some careless typographical errors in the text, such as PhD, PDL1 (PD-L1 in a table title), aPD-1, aPD1:PDL1, Janus kinase (after JAK2), and signal transducer and activation (after STAT).
